# Automated Detection of Surgical Implants on Plain Knee Radiographs Using a Deep Learning Algorithm

**DOI:** 10.3390/medicina58111677

**Published:** 2022-11-19

**Authors:** Back Kim, Do Weon Lee, Sanggyu Lee, Sunho Ko, Changwung Jo, Jaeseok Park, Byung Sun Choi, Aaron John Krych, Ayoosh Pareek, Hyuk-Soo Han, Du Hyun Ro

**Affiliations:** 1College of Medicine, Seoul National University, Seoul 03080, Republic of Korea; 2Department of Orthopedic Surgery, Korean Armed Forces Yangju Hospital, Yangju-si 11429, Republic of Korea; 3Department of Computer Science and Engineering, Seoul National University, Seoul 08826, Republic of Korea; 4Department of Orthopedic Surgery, Seoul National University Hospital, Seoul 03080, Republic of Korea; 5Seoul National University Hospital, Seoul 03080, Republic of Korea; 6Department of Orthopedic Surgery, Mayo Clinic, Rochester, MN 55902, USA; 7CONNECTEVE Co., Ltd., Seoul 03080, Republic of Korea

**Keywords:** automated detection, detection algorithm, deep learning

## Abstract

*Background and Objectives*: The number of patients who undergo multiple operations on a knee is increasing. The objective of this study was to develop a deep learning algorithm that could detect 17 different surgical implants on plain knee radiographs. *Materials and Methods*: An internal dataset consisted of 5206 plain knee antero-posterior X-rays from a single, tertiary institute for model development. An external set contained 238 X-rays from another tertiary institute. A total of 17 different types of implants including total knee arthroplasty, unicompartmental knee arthroplasty, plate, and screw were labeled. The internal dataset was approximately split into a train set, a validation set, and an internal test set at a ratio of 7:1:2. You Only look Once (YOLO) was selected as the detection network. Model performances with the validation set, internal test set, and external test set were compared. *Results*: Total accuracy, total sensitivity, total specificity value of the validation set, internal test set, and external test set were (0.978, 0.768, 0.999), (0.953, 0.810, 0.990), and (0.956, 0.493, 0.975), respectively. Means ± standard deviations (SDs) of diagonal components of confusion matrix for these three subsets were 0.858 ± 0.242, 0.852 ± 0.182, and 0.576 ± 0.312, respectively. True positive rate of total knee arthroplasty, the most dominant class of the dataset, was higher than 0.99 with internal subsets and 0.96 with an external test set. *Conclusion*: Implant identification on plain knee radiographs could be automated using a deep learning technique. The detection algorithm dealt with overlapping cases while maintaining high accuracy on total knee arthroplasty. This could be applied in future research that analyzes X-ray images with deep learning, which would help prompt decision-making in clinics.

## 1. Introduction

As knee operations are increasing in number, the number of patients who undergo multiple knee surgeries is increasing [1,2]. Surgical history must be identified to check whether there is an existing implant and what type of implant it is [3]. However, some patients undergo surgeries at multiple institutions, which hinders obtaining accurate surgical history [4]. According to a 2012 survey on American Association of Hip and Knee Surgeons (AAHKS) members, 10% of implants before operations and 2% of implants during surgery could not be identified [5]. Moreover, the median time taken for implant identification by surgeons was 20 min [5].

Automated identification using artificial intelligence (AI) is an efficient way to overcome this situation. Machine learning (ML) methods have been demonstrated to be fast and accurate in processing medical images such as X-ray, MRI, and pathology slides [6,7]. Among diverse ML approaches, deep learning (DL) especially stands out because it can find distinctive features in given data without direct human supervision, unlike classic ML models [8]. DL model could significantly reduce time and cost spent in identifying implants by examining X-ray images directly and automatically. Tiwari et al. [9] and Patel et al. [10] have directly compared DL models with human experts in examining X-ray images and proven that DL models are better in both speed and accuracy.

Ren et al. [11] have reviewed studies on identification of orthopedic implants using AI. Their systematic review covered several different anatomic sites such as hip, knee, and shoulder. It is reasonable because basic concepts are similar regardless of anatomical locations. Among the 11 studies they included, those conducted by Yi et al. [12] and Karnuta et al. [13] dealt with knee X-rays.

Yi et al. [12] were the first who tried to identify implants on knee plain radiographs using AI. They developed three different DL models: (1) a model that judges whether there is total knee arthroplasty (TKA) on the image, (2) a model that classifies TKA and unicompartmental knee arthroplasty (UKA), and (3) a model that distinguishes between two TKA models. They utilized 237 to 274 images, which varied by each model. All three models with binary classifications achieved a perfect AUC of 1.0.

Karnuta et al. [13] have expanded the task into multi-class classification. Their DL system classified nine different implant models with AUC of 0.99. They also increased the size of the dataset into 682 images.

Contrary to these previous studies [12,13], this study employed a detection network because there were ‘overlapping’ cases in the dataset. A majority of previous studies focused on classifying companies and models within the scope of TKA, whereas this study aimed to figure out the type and location of every implant on an X-ray image. For example, some images might include TKA and three screws simultaneously. Such images could be easily identified with detection networks.

Covering a variety of implants other than TKA is important because TKA is not the only surgical intervention that a patient may have undergone. For instance, if a patient received high tibial osteotomy (HTO), a plate and screws would appear on the X-ray, but there would be no TKA. Another example is anterior cruciate ligament (ACL) reconstruction. Usually, only screws would appear in this case. Most of the previous studies could not deal with these surgeries because their models classified TKA only. Thus, the main purpose of this study was to develop a DL model to identify 17 different types of objects and determine their locations in the form of bounding boxes.

## 2. Materials and Methods

This study was approved by the Institutional Review Board (No. H-1801-061-915). The overall process of this study, from organizing dataset to evaluating the trained DL model, is summarized in Figure 1.

### 2.1. Dataset and Pre-Processing

The internal dataset consisted of 5206 X-rays taken at a single, tertiary hospital from 4435 patients. The external test contained 238 X-rays from another tertiary institution. X-rays from each hospital were retrospectively collected from 2009-01-01 to 2019-12-31.

All images were knee anteroposterior (AP) view images. These images were first converted from Digital Imaging and Communications in Medicine (DICOM) format to Portable Network Graphics (PNG) format. They were subsequently converted into Joint Photographic Expert Group (JPG) format. JPG quality was 70 for internal data and 95 for external data.

The external dataset consisted of 71 images including both legs (29.8%), 73 images including only the left leg (30.7%), and 94 images including only the right leg (39.5%). Those images were flipped horizontally and concatenated with the original ones. If the original image contained the right leg, the flipped one was put left to it. If the original one included the left leg, the flipped one was placed right to it.

A total of 17 different types of objects were labeled: TKA_femur, TKA_tibia, UKA, staple, wire, screw, plate, IM nail, washer, smooth pin, metal button, bone cement, tumor prosthesis, ruler, External fixator, Patellofemoral arthroplasty, and TKA_stem (TKA = total knee arthroplasty, UKA = unicompartmental knee arthroplasty, IM nail = intramedullary nail). The type and location of each object were labeled so that both classification and detection were possible. Table 1 shows the number of images and instances that belong to each class in internal and external datasets.

The internal dataset was split into train set, validation set, and internal test set. The split ratio was train: validation: test = (about) 7:1:2. Multiple X-rays from a single patient were assigned to the same subset. In addition, the class imbalance was adjusted by applying a ratio of 7:1:2 in a per-class manner. Except for this principle, the splitting process was done in a random manner. Finally, 3701, 517, and 1042 images were assigned to the train set, validation set, and internal test set, respectively. The external dataset was solely used as the external test set. These 238 images were neither used in the training nor in the validation of DL process.

### 2.2. Deep Learning Model Training

You only look once (YOLO) was utilized as the DL algorithm. The detection approach before YOLO was divided into two steps. For example, R-CNN could “first generate potential bounding boxes in an image and then run a classifier on these proposed boxes” [14]. Using YOLO, the detection process became a single-step, “straight from image pixels to bounding box coordinates and class probabilities” [14]. In other words, it can judge the type and location of an object simultaneously. YOLO is fast and light because of this feature. Yolov5, the latest version of YOLO, was employed in this research. Yolov5 offers multiple options such as size and complexity, whose basic approaches are identical. Yolov5s has the most simple and lightest architecture among them. Thus, it was selected in this study.

The programming language and the deep learning framework used for model training were Python 3.8.5 and PyTorch, respectively. Graphics processing unit (GPU) NVIDIA GeForce RTX 3090 was utilized for fast training. Transfer learning was conducted. That is, Yolov5s trained with Common Objects in Context (COCO) dataset was fine-tuned to adjust to our dataset. COCO is a large-scale dataset frequently used in detection and segmentation tasks like ImageNet in classification tasks. It consists of 330K images with 1.5 million object instances labeled into 80 object categories [15]. Detection models pretrained with COCO is expected to perform well even on novel tasks because it has already learned how to find meaningful features on various kinds of objects (e.g., finding contours or rectangles). The model was trained and validated in 50 epochs. The batch size was 16. Online imagespace and colorspace augmentations were applied to the train set [16]. Three random images were presented with the original image each time when an image was loaded for training [17].

### 2.3. Evaluation Metrics

Train losses and validation losses of each epoch were examined to verify that no overfitting occurred. In that case, the model ‘memorizes the answers’ of the training set instead of recognizing genuine features of each image. This phenomenon can be sensed by tracking validation loss because an overfitted model cannot perform well on novel data. Thus, validation loss increases with each epoch in the overfitted model.

Accuracy, sensitivity, specificity, and confusion matrix were used to evaluate the model’s performance. These metrics were each measured with the validation set, internal test set, and external test set separately. Accuracy, sensitivity, and specificity of each class were first calculated. The total score was defined as their weighted average, whose weight was the number of instances that belonged to each class. Accuracy was defined as (TP+TN)/(TP+TN+FP+FN). Sensitivity was defined as TP/(TP+FN) and specificity was defined as TN/(TN+FP). The confusion matrix is a figure that shows TP, TN, FP, and FN rates of each class. TP, TN, FP, and FN mean true positive, true negative, false positive, and false negative, respectively. The metrics of each class could be figured out by treating the task like a binary classification problem. For instance, the sensitivity value of TKA_stem was figured out by defining TKA_stem as ‘True’ and all other classes as ‘False’. The judgment of true and false is intuitive in classification. For example, if the ground truth is ‘0’ and the model prediction is ‘0’, it is true. If the model prediction is ‘1’, it is false. However, in the case of detection, the location of the bounding box was also taken into consideration.

Intersection over union (IoU) was used as a criterion for detection. IoU was defined as (area of overlap intersection)/(area of union). It was determined by overlap intersection and union between two boxes: ground truth and prediction. IoU became higher when the two boxes overlapped more, i.e., the area of intersection of the two boxes was larger while the area of symmetric difference was smaller. It is because the area of union is the sum of the area of intersection and symmetric difference. The threshold of minimum IoU can be selected by users. In this study, a threshold of 0.5 was used for accuracy, sensitivity, and specificity, while a threshold of 0.45 was for confusion matrices. It means that if the threshold is 0.7, predictions with IoU lower than 0.7 are false predictions.

## 3. Results

Figure 2 shows no overfitting in the training process. There was no abnormal increase in validation loss during the task.

Table 2, Table 3 and Table 4 present accuracy, sensitivity, and specificity tested with the validation set, internal test set, and external test set, respectively. Total accuracy, total sensitivity, total specificity value of each subset were (0.978, 0.768, 0.999), (0.953, 0.810, 0.990), and (0.956, 0.493, 0.975), respectively.

As for the internal validation set, the accuracy ranged from 0.703 to 1. Accuracy values for TKA_tibia, UKA, bone cement, and tumor prosthesis were 1. The value for Patellofemoral arthroplasty was 0.703. Sensitivity values for TKA_tibia, UKA, metal button, bone cement, tumor prosthesis, and TKA_stem were 1. Specificity values for TKA_femur, TKA_tibia, UKA, wire, plate, bone cement, tumor prosthesis, External fixator, and Patellofemoral arthroplasty were also 1.

As for the internal test set, the accuracy ranged from 0.402 to 0.999. Accuracy values for UKA, bone cement, tumor prosthesis, ruler, and External fixator were 0.999. This value for Patellofemoral arthroplasty was 0.402. Sensitivity values for UKA, bone cement, and tumor prosthesis were 1. The specificity value for External fixator was also 1.

As for the external test set, the accuracy ranged from 0.670 to 0.998. The accuracy value for External fixator was 0.998. This value for screw was 0.670. Sensitivity values for TKA_femur and UKA were 1. Specificity values for UKA, smooth pin, and External fixator were also 1.

Confusion matrices of the validation set, internal test set, and external test set are displayed in Figure 3. Means ± SDs of diagonal components of the validation set, internal test set, and external test set in the confusion matrices were 0.858 ± 0.242, 0.852 ± 0.182, and 0.576 ± 0.312, respectively.

## 4. Discussion

The most dominant orthopedic implant of our dataset was TKA. This is because the number of arthritis patients is increasing due to the aging population. In addition, TKA has been acknowledged as a highly effective treatment, especially for senior patients. Nevertheless, there are other types of implants for other diseases. The number of these cases was not negligible. Thus, the authors tried to define as diverse classes as possible to cover a wide range of knee implants.

Table 5 compares the approach and the result of this research with previous studies. Among previous studies that automatically identified implants on X-rays, seven studies that dealt with knee radiographs were included. In comparison, our study dealt with large amount of dataset with comparable accuracy to other DL models while adapting a different kind of approach (detection method) to identify multiple different kinds of implants.

Belete et al. [18] have developed a DL model with eight different classes, and they considered calculating the probability with the softmax function in the evaluation process. Sharma et al. [19] have classified six implant types using 1078 images and compared four different networks with an external test set that consists of 162 images. Tiwari et al. [9] have classified six implant types using 521 images, and they have shown the superiority of ML models over human experts regarding the average accuracy. Patel et al. [10] have used both knee and hip radiographs, identifying 12 implant types using 427 knee and 922 hip unilateral images. Klemt et al. [20] have also classified both knee and hip arthroplasties, identifying 14 TKA designs and 24 total hip arthroplasty (THA) designs using a total of 11,204 knee X-rays.

Gurung et al. [21] have reviewed nine studies using AI for the identification of orthopedic implants. They also analyzed three articles that focused on predicting the risk of implant surgery failure. Ren et al. [11] and Gurung et al. [21] both pointed out that most of the existing studies didn’t use external datasets, which means they only used radiographs from one institution. Our model was tested on a balanced external test set to resolve this problem.

Among studies that only handled locations other than the knee, such as the hip and shoulder, the one conducted by Kang et al. [22] was notable because they automatically cropped images with YOLOv3, a detection network. The final prediction was done with a simple CNN classification network. The research by Urban et al. [23] was also noticeable because they compared the performances of DL networks with other ML classifiers (e.g., Gradient Boosting).

The current study employed YOLO, a detection network, unlike majority of previous studies that utilized classification networks such as ResNet. These previous studies have focused on identifying different companies or models of arthroplasty implants like TKA because different models of arthroplasties are usually incompatible. In contrast, this study aims to identify implants that can appear simultaneously in the same image. In other words, images that include each type of object are not disjoint in our dataset. For example, both UKA and screw can appear in some images. Therefore, when applying classification networks, this task becomes not only a multi-class classification, but also a multi-label classification. To handle this problem, either the dataset or model algorithm should be transformed. That was why this study employed detection approach instead. Unlike classification, detection networks can inherently handle overlapping cases. The model output includes every object in an image. For example, the model can detect eight smooth pins and two wires in an image and figure out their locations.

As to the pre-processing of images, conversion into low-quality JPGs can reduce not only computational resources required for DL training and inference, but also model training and predicting time. Another way to save time and resources is by cropping images to only include the region of interest (ROI), i.e., the area around the knee joint. However, cropped dataset limits types and locations of implants that can be detected. For example, detection of TKA_stem and differentiation between IM_nail and TKA_stem becomes easier with the dataset that is uncropped. Thus, the whole image was used in this study.

In terms of defining object classes, the reason why TKA was split into TKA_femur and TKA_tibia and why there were classes that were not surgical implants (i.e., ruler and external fixator) was related to the performance of the detection model. TKA was split into TKA_femur and TKA_tibia because unless it was separated, bounding boxes became too large and the performance of the detection model would decline. The reason why there were ‘ruler’ and ‘external fixator’ classes was because unless they were assigned to some classes, the detection model could mistake them for some other implants like screws or staples.

As for model evaluation, the metrics were measured with three subsets for comparison. Test sets were more novel than the validation set to the model. The external test set was from different institution. Thus, how the model would respond in a new situation can be predicted by comparing its performances with these three subsets. Moreover, the reason why the ‘weighted’ average of accuracy, sensitivity, and specificity was calculated was because it could help deal with the class imbalance problem by reflecting the number of instances. Despite covering diverse cases, the proportion of TKA instances was high, which was more than half of the entire dataset.

Confusion matrices in Figure 3 showed that the model was more than 99% accurate on TKA in the case of internal subsets. This score was almost the same as the binary TKA classifier described by Yi et al. [12]. TP rate of TKA on external subset was also high, which was about 0.96. The fact TKA was the most dominant class might explain this result. However, this result might also be due to fact that TKA implant is relatively big and easy to identify owing to its distinct shape. UKA shares these characteristics with TKA. Its TP rates were 1.00 in all subsets. Interestingly, TP rates of screw, the second most dominant class after TKA, were less than 0.9 in all three subsets. It might be because the model confused screws with other small implants like staples.

External fixator was not detected successfully because the number of their instances was too small. In the case of tumor prosthesis whose number of instances was also small, the model showed perfect TP rates with internal subsets, but a poor TP rate of 0.3 with the external dataset. It means the model is not robust enough for detecting tumor prosthesis. Notably, the score on patellofemoral arthroplasty was fine, although its number of instances was similar to that of tumor prosthesis. The fact that its shape and location are very distinct from other implants might explain this outcome.

The metrics in Table 2, Table 3 and Table 4 display similar tendencies with the confusion matrices. Performances for TKA and UKA were high in all three subsets. The model could successfully identify the presence of TKA and UKA. Overall scores with the external dataset show that the robustness of the model needs to be improved. However, the evenness of the external test set should be considered, which can be verified through the number of images by class such as those shown in Table 1.

This research would be helpful not only in clinical fields including medical centers performing orthopedic surgeries, but also in future DL studies. Inclusion or exclusion criteria for DL research on X-rays may contain whether a patient has undergone knee surgeries. Automated identification of implants would help us quickly screen images for implants.

This study has several limitations. First, there was a class imbalance problem in the internal dataset. The proportion of TKA instances was very high, which was more than half of the entire dataset. It can make an objective evaluation of the DL models difficult. This problem could be partly resolved by using a relatively balanced external dataset. Second, the sensitivity of the model for certain kinds of implants such as staples was low. Since this was the first study, to our knowledge, that dealt with these low profile implants, further study is needed to improve the sensitivity of the model. Third, default hyperparameters were used for model training. The model performance could become better if hyperparameter tuning was applied.

## 5. Conclusions

In conclusion, this study explored automation of implant identification process on plain knee radiographs using DL techniques. True positive rates of TKA, the most dominant class, for the validation set, internal test set, and external test set were 1.00, 0.99, and 0.95, respectively. The total accuracy value of the internal test set was 0.953, meaning that the model classified other classes with a fine performance as well. Unlike the models presented in previous studies, the one in this study can identify various cases other than TKA in clinical fields by detecting multiple implants that can be overlapped. Approaches of this study can also be applied in future research that analyzes X-ray images with DL techniques.

## Figures and Tables

**Figure 1 medicina-58-01677-f001:**
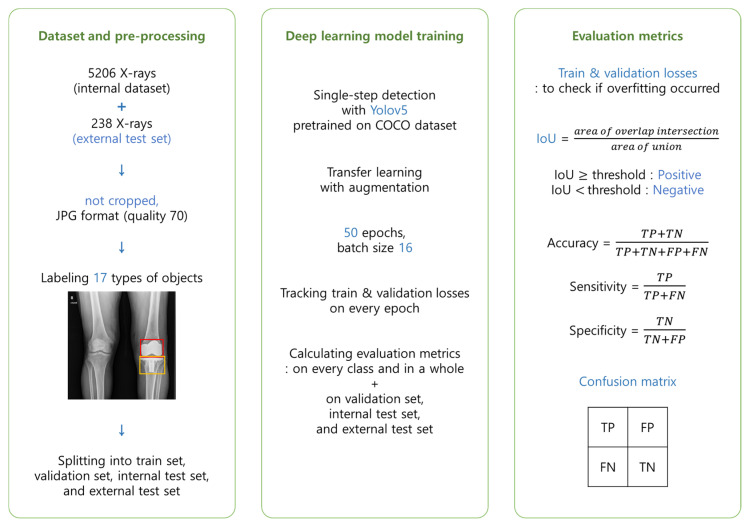
A schematic process of the deep learning system development in this study. (TP = true positive, TN = true negative, FP = false positive, FN = false negative, mAP = mean average precision).

**Figure 2 medicina-58-01677-f002:**
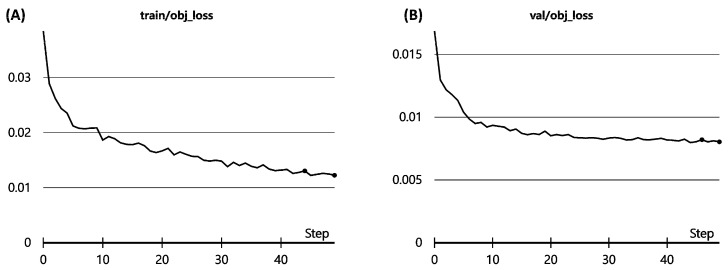
Train (**A**) & validation (**B**) losses of the Yolov5s model in the study. As the number of epochs increases, train losses always decrease. Validation losses decrease if the model is robust on new data. If the model is overfitted to the train set, validation losses would increase.

**Figure 3 medicina-58-01677-f003:**
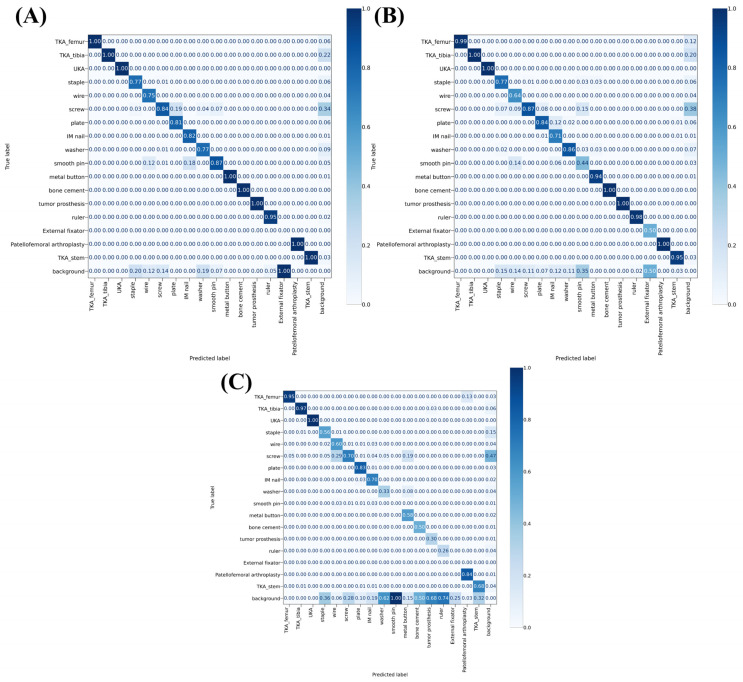
Confusion matrices on validation set (**A**), internal test set (**B**), and external test set (**C**). Among 18 labels on each axis, only ‘background’ is not an object class. If the ‘True label’ is background and the ‘Predicted label’ belongs to one of the object classes, it means that the model made a false positive judgment on the presence of an implant. If the ‘Predicted label’ is background and the ‘True label’ belongs to one of the object classes, it means that the model made a false negative judgment on the presence of an implant.

**Table 1 medicina-58-01677-t001:** The number of images and instances by class in internal and external dataset.

	The Internal Dataset		The External Dataset	
Class	Number of Images	Number of Object Instances	Number of Images	Number of Object Instances
TKA_femur	3913	6107	46	85
TKA_tibia	3911	6101	41	76
UKA	154	173	21	41
staple	189	310	23	55
wire	89	126	26	68
screw	1035	3348	113	1184
plate	283	371	65	151
IM nail	83	89	40	79
washer	200	245	19	42
smooth pin	62	170	7	12
metal button	155	204	29	99
bone cement	11	11	11	20
tumor prosthesis	12	12	20	37
ruler	237	237	16	31
External fixator	6	10	1	4
Patellofemoral arthroplasty	10	13	22	31
TKA_stem	310	521	10	25

TKA, total knee arthroplasty; UKA, unicompartmental knee arthroplasty; IM, intramedullary.

**Table 2 medicina-58-01677-t002:** Evaluation metrics on the validation set.

	Accuracy	Sensitivity	Specificity
TKA_femur	0.999	0.998	1
TKA_tibia	1	1	1
UKA	1	1	1
staple	0.991	0.6	0.999
wire	0.999	0.75	1
screw	0.952	0.737	0.996
plate	0.997	0.806	1
IM nail	0.998	0.818	0.999
washer	0.994	0.654	0.999
smooth pin	0.997	0.733	0.999
metal button	0.999	1	0.999
bone cement	1	1	1
tumor prosthesis	1	1	1
ruler	0.999	0.952	0.999
External fixator	0.999	0	1
Patellofemoral arthroplasty	0.703	0.002	1
TKA_stem	0.999	1	0.999
**Total**	**0.978**	**0.768**	**0.999**

TKA, total knee arthroplasty; UKA, unicompartmental knee arthroplasty; IM, intramedullary.

**Table 3 medicina-58-01677-t003:** Evaluation metrics on the internal test set.

	Accuracy	Sensitivity	Specificity
TKA_femur	0.998	0.998	0.998
TKA_tibia	0.994	0.997	0.985
UKA	0.999	1	0.999
staple	0.979	0.754	0.987
wire	0.995	0.818	0.998
screw	0.888	0.868	0.901
plate	0.99	0.92	0.993
IM nail	0.994	0.647	0.997
washer	0.987	0.864	0.99
smooth pin	0.984	0.5	0.994
metal button	0.995	0.971	0.996
bone cement	0.999	1	0.999
tumor prosthesis	0.999	1	0.999
ruler	0.999	0.979	0.999
External fixator	0.999	0.5	1
Patellofemoral arthroplasty	0.402	0.002	0.999
TKA_stem	0.993	0.948	0.996
**Total**	**0.953**	**0.810**	**0.990**

TKA, total knee arthroplasty; UKA, unicompartmental knee arthroplasty; IM, intramedullary.

**Table 4 medicina-58-01677-t004:** Evaluation metrics on the external test set.

	Accuracy	Sensitivity	Specificity
TKA_femur	0.988	1	0.987
TKA_tibia	0.981	0.987	0.981
UKA	1	1	1
staple	0.944	0.527	0.958
wire	0.975	0.588	0.99
screw	0.67	0.644	0.725
plate	0.982	0.848	0.995
IM nail	0.979	0.658	0.994
washer	0.974	0.238	0.992
smooth pin	0.993	0	1
metal button	0.966	0.505	0.994
bone cement	0.989	0.35	0.997
tumor prosthesis	0.979	0.243	0.995
ruler	0.979	0.161	0.994
External fixator	0.998	0	1
Patellofemoral arthroplasty	0.878	0.118	0.999
TKA_stem	0.972	0.52	0.979
**Total**	**0.956**	**0.493**	**0.975**

TKA, total knee arthroplasty; UKA, unicompartmental knee arthroplasty; IM, intramedullary.

**Table 5 medicina-58-01677-t005:** Comparison with previous studies.

Author	Year	Total Dataset Size	External Test Set	Radiograph Location	Mode of Judgement	Accuracy
Yi et al. [12]	2020	511	No	knee	classification	Not reported (AUC = 1.00)
Karnuta et al. [13]	2021	682	No	knee	classification	99%
Belete et al. [18]	2021	588	No	knee	classification	100%
Sharma et al. [19]	2021	1240	Yes	knee	classification	96%
Tiwari et al. [9]	2022	521	No	knee	classification	96%
Patel et al. [10]	2021	1547	No	knee and hip	segmentation and classification	99%
Klemt et al. [20]	2020	11204	No	knee and hip	classification	Not reported (AUC = 0.97)
Kim et al. (This study)	2022	5444	Yes	knee	detection	96%

AUC = Area Under the Curve.

## Data Availability

The data presented in this study are available on request from the corresponding author. The data are not publicly available due to privacy.

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
