# Peer review of "Automated Detection of Surgical Implants on Plain Knee Radiographs Using a Deep Learning Algorithm"

_medicina, 2022, doi:10.3390/medicina58111677_

Round 1

Reviewer 1 Report (Previous Reviewer 1)

Dear Authors, 

thank you for the chance to review this manuscript. 
The manuscript is well written.

You  will find some view comments to improve your manuscript for each section below.

1. Please shorten the introduction: The need for AI approaches and deep learning approaches becomes clear, basically without the long section  from L.71-L.111. 
2. Please stay objective in the introduction, yes there is a need, but it is not the only way of getting more sufficient.

Methods:
139-140: Please report how many out of how many have been reported for both legs (+ percentage)

Please write methods in passive voice (e.g. L 140-141)

Results: well written and shown

Discussion: Please extend discussion on benefits normal PPV, Accuracy of other approaches

Author Response

Reviewer 2 Report (Previous Reviewer 2)

• For “2.3 Evaluation metrics”, a flow-chart can be drawn to make it easier to understand.

• The texts in Figure 3 should be made visible, and the shape should be improved.

• The superiority of the study should be emphasized by creating a table in which the results obtained from the current study and the results obtained in the literature are compared.

Author Response

This manuscript is a resubmission of an earlier submission. The following is a list of the peer review reports and author responses from that submission.

Round 1

Reviewer 1 Report

The conclusions must be rewritten A sensitivity of .78 is for sure not enough for implementation in clinical pratice. Is the review board the mocal ethic comittee? Because if those conclusion are implemented it might has wrong consequencies for patients.

Reviewer 2 Report

• The extended version of TKA (total knee arthroplasty) should be given in the abstract and in the first place in the text.

• The importance of the current work should be emphasized in the introduction section. It should be explained which gap the study will fill in the literature. It should be mentioned why this study is needed.

• Long versions of abbreviations such as TP can be added to the Figure 1 caption.

• Instead of “with” in line 190, “while” should be written.

• For “2.3 Evaluation metrics”, a flow-chart can be drawn to make it easier to understand.

• Figure 2 description can be expanded. Adjusting the hue in Figure 2 can make it black, and the axes can be visible.

• Figure 3 description can be expanded. The texts in Figure 3 should be made visible, and the shape should be improved.

• The superiority of the study should be emphasized by creating a table in which the results obtained from the current study and the results obtained in the literature are compared.

• How the current work sheds light on future research and application studies and its superiority over the literature should be briefly mentioned at the end of the conclusion.

• References from recent years should be increased by citing appropriate places. (Especially 2022)